# Relationship between Behavioral and Objective Measures of Sound Intensity in Normal-Hearing Listeners and Hearing-Aid Users: A Pilot Study

**DOI:** 10.3390/brainsci12030392

**Published:** 2022-03-15

**Authors:** Elsa Legris, John Galvin, Yassine Mofid, Nadia Aguillon-Hernandez, Sylvie Roux, Jean-Marie Aoustin, Marie Gomot, David Bakhos

**Affiliations:** 1UMR-S1253, Université François-Rabelais de Tours, Inserm, 37000 Tours, France; jg3drop@gmail.com (J.G.); yassine.mofid@univ-tours.fr (Y.M.); nadia.aguillon-hernandez@univ-tours.fr (N.A.-H.); sylvie.roux@univ-tours.fr (S.R.); marie.gomot@univ-tours.fr (M.G.); david.bakhos@univ-tours.fr (D.B.); 2Ear Nose and Throat Department, CHRU de Tours, 37000 Tours, France; jm.aoustin@audilab.fr; 3House Institute Foundation, Los Angeles, CA 90057, USA

**Keywords:** loudness perception, hearing aid, cortical auditory evoked potential, pupillometry, objective measures

## Abstract

Background: For hearing-impaired individuals, hearing aids are clinically fit according to subjective measures of threshold and loudness. The goal of this study was to evaluate objective measures of loudness perception that might benefit hearing aid fitting. Method: Seventeen adult hearing aid users and 17 normal-hearing adults participated in the study. Outcome measures including categorical loudness scaling, cortical auditory evoked potentials (CAEPs), and pupillometry. Stimuli were 1-kHz tone bursts presented at 40, 60, and 80 dBA. Results: Categorical loudness scaling showed that loudness significantly increased with intensity for all participants (*p* < 0.05). For CAEPs, high intensity was associated with greater P1, N1, and P2 peak amplitude for all listeners (*p* < 0.05); a significant but small effect of hearing aid amplification was observed. For all participants, pupillometry showed significant effects of high intensity on pupil dilation (*p* < 0.05); there was no significant effect of hearing aid amplification. A Focused Principal Component analysis revealed significant correlations between subjective loudness and some of the objective measures. Conclusion: The present data suggest that intensity had a significant impact on loudness perception, CAEPs, and pupil response. The correlations suggest that pupillometry and/or CAEPs may be useful in determining comfortable amplification for hearing aids.

## 1. Introduction

Hearing-impaired individuals experience decreased auditory dynamic range due to recruitment and the loss/dysfunction of outer hair cells [1]. A dysfunction of outer hair cells reduces cochlear mechanical amplification of low-intensity sounds without altering cochlear mechanical responses to high-intensity sounds [2,3]. Approximately 60–70% of hearing loss can be explained by a loss of cochlear amplification [4,5]. To restore amplification, hearing-impaired individuals can use hearing aids (HAs), which are fit according to individual patterns of hearing loss, as reflected by the audiogram, with compression applied to increase the auditory dynamic range [6,7]. However, these clinical fitting methods do not guarantee patient satisfaction with their HAs. Patients may need several adjustments to their hearing aids before being satisfied with the fittings [8,9,10,11,12,13].

While HAs are clinically fit using subjective measures of threshold and comfortable loudness, some individuals may not be able to provide subjective judgements of loudness due to difficulties in communication [14]. For very young children, proper HA fitting is essential for successful outcomes. Early rehabilitation is recognized as a prognostic factor for good language development [15,16,17,18]. It is essential that threshold levels are not set too low (to avoid information loss) and comfort levels are not set too high (to avoid discomfort) during fitting to improve HA uptake and reduce HA rejection by patients. For adults, early diagnosis is also essential for good outcomes. Indeed, late rehabilitation could lead to the absence of cortical reorganization [19]. As hearing-impaired patients will receive new sound information from their device, it is important that HA fitting achieves auditory comfort. In clinical fitting of HAs, measuring hearing thresholds, comfort levels, and discomfort levels require patient participation. These subjective measures may be difficult for adult patients with other functional deficits (e.g., paralysis, cognition, etc.) and even more difficult in very young children (e.g., <12 months old) given the novelty of hearing for the first time.

Currently, there are few objective measures available to quantify auditory comfort levels, which might improve patient satisfaction with HA fitting. Cortical auditory evoked potentials (CAEPs) have been used as a clinical tool to fit HAs for infants [20,21]. CAEPs may be useful for young children and for difficult populations where behavioral responses are limited and/or unreliable [22]. CAEPs provide information regarding auditory stimulation at the cortical level. The main CAEP responses of interest are the two positive waves, P1 (at 60–80 ms) and P2 (at 100–160 ms), and the negative peak N1 (at 90–100 ms) [23]. However, recording cortical responses is too long and tedious to be used in clinical practice. Stapedial reflex can be used to estimate the upper limit of loudness [24,25] but cannot be used to estimate comfortable loudness levels.

Pupillometry is an objective measure that has been used to estimate pain [26,27], measures of memory load [28,29], selective attention [30], motivation [31], and linguistic coherence of stimuli [32]. Increased pupil dilation reflects increased deployment of the sympathetic branch of the autonomic nervous system [33]. Pupillary constriction and dilation are mediated by autonomic regulation of the circular and radial fibers of the iris [26,27,34]. Variations in pupil diameter are influenced by the autonomic nervous system, which controls regulatory functions to adapt to environmental demands [35]. Pupil dilation could reflect a reaction to an endogenous or an exogenous stimulus. Some recent studies have used pupillometry as an objective tool to evaluate listening effort [29,36,37,38]. Pupil dilation can reflect many things at the same time, including participants’ response to a task, their momentary state of mind (i.e., emotional and attentional state), and their cognitive capacity [39,40]. In addition, greater variability in the morphology of pupil dilation curves has been observed in cochlear implant users, compared to normal hearing (NH) listeners, suggesting that degraded auditory input may affect pupil responses [41].

Pupillometry has been used to show the efficacy of HAs. Bianchi et al. (2019) showed that bone-anchored hearing aids (BAHAs) with a higher maximum force output provided better speech understanding in noise, which was reflected by lower pupil dilation, suggesting less reduced effort [42]. Similarly, activation of HA noise reduction resulted in reduced listening effort [43], better performance, and smaller pupil dilation at low input signal-to-noise ratios (SNRs) [44]. CAEPs have been used to highlight how signal processing (microphone directionality or noise reduction) influences neural activity in the brain for individuals with hearing loss [45]. Taken together, these studies underline the potential for pupillometry and CAEPs to objectively demonstrate the impact of HA signal processing (e.g., maximum force output, microphone directionality or noise reduction). 

Previous studies have shown that NH listeners exhibit larger pupil responses with increasing intensity [46,47,48], as well as shorter peak latencies and higher amplitudes for CAEP responses [49,50,51,52,53,54,55,56,57,58,59,60,61,62]. Other previous studies showed that HA amplification reduced CAEP amplitude [63] or had no effect on CAEP responses [62,64]. To our knowledge, no studies have explored pupil responses and CAEPs as a function of sound intensity for individuals with hearing loss or who use HAs. While pupil dilation and EEG recordings have been widely used to detect changes in listening effort, they have not been studied together in response to changes in sound intensity [43]. The primary objective of this study was to analyze how pupillometry and CAEP are influenced by sound intensity for NH listeners and individuals with hearing loss that use HAs. The secondary objective was to analyze subjective loudness perception, which is well known to be linked to sound intensity, and to observe potential associations between behavioral and objective measures. If associations were to be found, objective measures may predict auditory comfort or discomfort, and could be used in clinical HA fitting.

## 2. Methods

### 2.1. Participants

Seventeen adult (9 women, 8 men), right-handed, native French speakers with presbycusis and moderate sensorineural hearing loss (SNHL) according to Bureau International d’Audiophonologie (BIAP) criteria [65] participated in the study. All hearing-impaired participants were bilateral hearing aid (HA) users for at least one year and wore their HA at least 6 h per day; the mean daily HA use was 10.0 ± 3.2 h according to data logging. Participants were recruited from the Cochlear Implant unit of the Otolaryngology Department at University Hospital of Tours, France. None of the participants had reported neurologic disorders or history of Meniere’s disease. The mean age at testing was 76.0 ± 7.0 years (range: 62–88 years), and all had a mini mental state score of 30/30. Throughout this study, these participants are referred to as the “HA group”. Table 1 shows demographic information for the HA group. The mean unaided air pure tone average (PTA) threshold across 0.5, 1.0, 2.0, and 4.0 kHz was 47.7 ± 14.0 dB HL for left ear and 47.5 ± 11.7 dB HL for right ear; a paired t-test showed no significant difference in PTA thresholds between the right and left ear (*p* = 0.902). The unaided speech audibility threshold (SAT) for French monosyllable words [66] was 52.9 ± 12.5 dB HL for the left ear and 51.7 ± 7.4 dB HL for the right ear; a paired t-test showed no significant difference in SATs between the right and left ear (*p* = 0.627). The mean amplification for PTA thresholds (the difference between aided and unaided PTA thresholds across both ears, evaluated in the sound field) was 16.4 ± 8.6 dB HL. The mean amplification for SATs (the difference between aided and unaided SATs across both ears, evaluated in the sound field) was 12.5 ± 8.0 dB SPL.

Seventeen normal hearing (NH) adults (11 women, 6 men) served as experimental controls. All had PTA thresholds ≤25 dB HL and none had any reported neuronal disease. The mean age at testing was 64.0 ± 3.9 years (range: 59–75 years), and all had a mini mental state score of 30/30. Throughout the study, these participants are referred to as the “NH group”.

The Ethics Committee of the University Hospital of Tours specifically approved the protocol for participants in the HA group (N°ID RCB No. 2015-A01249-40) and the NH group (N°ID RCB No. 2017-A00756-47). Written informed consent was obtained from all participants.

### 2.2. Hearing Aids

All participants in the HA group used in-the-ear receiver HAs. All HA devices were less than 5 years old from Oticon (*n* = 10) or Bernafon (*n* = 7). HAs were fitted using a receiver of 85 dB and all had the same frequency range (8 kHz) and maximal gain (55 dB), as measured using a 2-cc coupler. HAs were programmed for omnidirectional amplification with a deactivated volume control. HAs were fit using NOAH software and the manufacturer’s programming module, and all were fit using the NAL NL2 prescription [6]. The HA gain was checked using a real ear insertion measure (REIR); an international speech test signal (ISTS) was presented at 65 dB SPL at 45° azimuth and recorded from the ear canal with the probe-tube microphone of the Affinity module to check that the gain was consistent with hearing loss.

### 2.3. Procedure

All tests were conducted in a single session which lasted an average of 65 min for the NH group and 100 min for the HA group, where tests were conducted with and without their HAs. The order of testing was EEG recording (15 min), followed by pupillometry (15 min), and then loudness rating (5 min). Between EEG recording and pupillometry, participants were given a 30-min break. If a participant felt asleep during recording, the test was stopped and a break was given before continuing the session.

#### 2.3.1. Categorical Loudness Scaling

Three 1-kHz tone bursts (total duration = 1 s) at three intensities (40, 60, 80 dBA) were presented in random order via two loudspeakers positioned 1.3 m away from the participant and at −45° and +45° relative to center. After the stimulus presentation, participants were asked to rate the loudness according to a scale that ranged from 0 (inaudible level) to 10 (intolerable level). Aided and unaided (with and without HA) loudness ratings were obtained from the HA group. For all participants, loudness ratings were obtained for each intensity 3 times.

#### 2.3.2. Cortical Auditory Evoked Potentials (CAEPs)

##### Stimuli

Stimuli (*n* = 360) were 1-kHz tone bursts (duration = 200 ms) generated using MATLAB (Mathworks, Natick, MA). Stimuli were presented at 40 (*n* = 120), 60 (*n* = 120), and 80 (*n* = 120) dBA via two loudspeakers situated at 1.3 m away from the participant and −45° and +45° relative to center. The inter-stimulus interval (ISI) varied from 2 to 3 s (offset to onset). For the HA group, cortical recordings were made with and without the HAs in two separate sessions. The orders of the stimulus intensity and listening condition (with or without HA) were randomized. The neurophysiological recordings took approximately 30 min for each session. During the recording session, participants sat in a sound-attenuated room and watched a silent movie.

##### Electroencephalogram (EEG) Data Recording

EEG data were recorded using a Compumedics System Neuroscan EEG system (Synamps RT amplifier and Curry 7 software) with 64 electrodes referenced on line to the nose. All electrodes were placed according to the international 10–20 electrode placement standard. Electrode impedances were kept below 5 kΩ. In addition, electrooculogram (EOG) activity was recorded from electrodes placed at the outer canthi of both eyes (horizontal EOG) and above and below the right eye (vertical EOG). The EEG data were recorded with a sampling frequency of 500 Hz and low-pass filtered at 200 Hz. The stimulus presentation was controlled by Presentation software.

EEG analysis was performed using ELAN software [67]. EEG recordings were filtered by a band-pass filter (0.3–70 Hz). Artifacts resulting from eye movements were removed using independent component analysis, and movement artifacts characterized by a high-frequency or high-amplitude signal were discarded manually by the experimenter. Afterwards, EEG was segmented into epochs from −100 to 500 ms relative to the stimulus onset. The epochs were baseline-corrected relative to a 100-ms pre-stimulus time window, and a digital zero-phase-shift low-pass filter of 30 Hz was applied. The mean number of epochs varied: 81.7 ± 14.6 (without HA), 82.3 ± 13.7 (with HA), and 85.6 ± 12.5 (NH) at 40 dBA; 80.8 ± 16.0 (without HA), 81.6 ± 13.6 (with HA), and 83.5 ± 12.9 (NH) at 60 dBA; and 77.5 ± 17.7 (without HA), 80.8 ± 14.3 (with HA), and 84.4 ± 12.7 (NH) at 80 dBA.

### 2.4. Pupillometry

#### 2.4.1. Stimuli

Stimuli were 1-kHz tone bursts (duration = 1 s) generated using MATLAB (Mathworks, Natick, MA, USA). Seven stimuli for each of the 40, 60, and 80 dBA intensities were presented via 2 loudspeakers situated at 1.3 m away from the subject and −45° and +45° relative to center. The order of intensity presentation was randomized, and the ISI was 20 ± 30 s (offset to onset). Pupillometry was measured with and without HAs in the HA group in two separate sessions.

#### 2.4.2. Data Recording

Visual stimuli were sent by an SMI iView X RED (version 2.8) remote eye-tracking system with a spatial resolution of 0.03° and a sampling frequency of 500 Hz. The system consisted of a computer equipped with two cameras sensitive to infrared light as well as a light source. A PC screen was positioned on top of the pupillometry system, about 45 cm away from the participant’s head. The illumination of the room (20 lux) was kept constant during the experiment for all participants. No eye-tracking equipment was needed to be worn by participants, as the corneal reflection of infrared light allowed for monitoring of ocular behavior.

The prototype of the experiment is presented in Figure 1. First, a white image was presented for 2 s, followed by a black image for 2 s, and finally a white image for 2 s to record the photo–motor response. For the rest of the experimental run, the visual stimulation was a gray image (20 lux) and included a cross to direct participants’ gaze on the screen. Experimental data recording began after 2 s of silence. Participants were seated in a comfortable armchair in a silent room and were asked to keep calm, not move, and watch the screen. For the HA group, pupillometry was measured with and without HAs.

### 2.5. Analysis

The baseline pupil size was determined as the average pupil size in the 1.0 s interval preceding the auditory stimulation. Only pupil sizes between 1 and 9 mm were considered for the analysis [68]. All remaining traces were baseline corrected by subtracting a baseline value from each time point within that trace. The mean pupil diameter at each intensity presentation was calculated by averaging the pupil diameter between stimulus onset and 5 s after stimulus offset. The mean pupil response at each intensity presentation was estimate during a period of 5 s from baseline (1 s before to 4 s after stimulus onset). If the pupil data contained more than 50% blinks between the start of the baseline and the prompt signal, it was excluded from the analysis. After this, the mean curves for each stimulation level were generated using MATLAB. Two pupil measures were extracted from the average trace: (1) peak dilation amplitude, defined as the maximum pupil diameter after the onset of the tone burst (peak level–baseline); and (2) latency of the peak dilation amplitude. Peak dilation amplitude was determined manually.

Pupil analysis data were analyzed for only 15 participants in the HA group, as data from two participants (HA-8 aided and HA-5 unaided) could not be used. For categorical loudness ratings and CAEP analysis, aided and unaided data were used for all 17 participants in the HA group.

#### Statistical Analyses

Analyses of variance (ANOVAs) were performed to evaluate the effects of intensity (40, 60, 80 dBA) and listening condition (NH, HA aided, HA unaided) on subjective and objective data. Within the NH and HA groups, separate repeated-measures (RM) ANOVAs were used to evaluate intensity effects (both groups) and HA effect (HA group); in cases where assumptions of normal distribution and equal variance were violated, non-parametric tests were used. Across the NH and HA groups, non-parametric tests were used to compare NH data to HA data (aided or unaided). For all analyses, the significance level was 0.05; Bonferroni or Tukey adjustments to the significance level were applied to all post hoc pairwise comparisons. Analyses were performed using IBM SPSS software.

Focused Principal Component Analysis (FCPA) was used to characterize relationships among the behavioral and objective measures. [69]. FCPA is based on Principal Component Analysis (PCA) and converts the structure of a correlation matrix into a distance matrix. FPCA allows for a graphical representation of associations between the dependent variable (here, subjective loudness scaling) and explanatory variables (here, objective measures of pupillometry and CAEPs), as well as the relationships among the explanatory variables [69,70]. FCPA was performed using the Psy library in R software, and figures were generated using the coorplot package in R software.

All data are reported in Appendix A.

## 3. Results

### 3.1. Categorical Loudness Scaling

Figure 2 shows boxplots of loudness ratings for the NH and HA groups (aided or unaided). In general, lo bSudness ratings increased with intensity for both groups. For the NH group, mean loudness ratings were 2.9 ± 1.7, 5.4 ± 1.5, and 7.8 ± 1.4 at 40, 60, and 80 dBA, respectively. A RM ANOVA was performed on the NH data, with intensity (40, 60, 80 dB) as the factor. Results showed a significant effect of intensity [F (2,32) = 124.0, *p* < 0.001]; Bonferroni-adjusted post hoc pairwise comparisons showed significant differences among all three intensities (*p* < 0.001 in all cases). For the HA group, mean loudness ratings with the HA off were 1.8 ± 1.4, 4.3 ± 2.3, and 7.2 ± 1.4 at 40, 60, and 80 dB, respectively; mean ratings with the HA on were 3.4 ± 1.3, 5.5 ± 1.1, and 7.2 ± 1.3 at 40, 60, and 80 dB, respectively. An RM ANOVA was performed on the HA group data, with HA (on, off) and intensity (40, 60, 80 dB) as factors. Results showed significant effects of HA [F (1,32) = 16.3, *p* < 0.001] and intensity [F (2,32) = 77.7, *p* < 0.001]; there was a significant interaction [F (2,32) = 11.0, *p* < 0.001]. Bonferroni-adjusted post hoc pairwise comparisons showed significant differences among all intensities for both listening conditions (*p* < 0.001 in all cases), and significantly higher ratings with the HA on than off at 40 and 60 dB (*p* < 0.001 in both cases), but not at 80 dB.

To determine across-group differences, separate Kruskal–Wallis ANOVAs on ranked loudness rating data were performed at each intensity, with listening group (NH, HA aided, and HA unaided) as the factor. Results showed significant effects of listening group at 40 (dF = 2, H = 11.1, *p* = 0.004) and 60 dB (dF = 2, H = 7.4, *p* = 0.024); there was no significant effect at 80 dB. After Bonferroni adjustment, post hoc Dunn pairwise comparisons showed no significant difference between the NH and the HA-aided or HA-unaided loudness ratings.

### 3.2. CAEPs

Figure 3 shows mean CAEP data for the NH and HA groups (aided and unaided). In general, peak amplitude increased with intensity for GFP and at Cz, and latency reduced with intensity. For the HA group, amplitude was generally higher and latency was generally earlier with the HA on than with the HA off. Amplitude and latency values were generally similar between the NH and HA group with the HA on, with slightly lower amplitudes and longer latencies observed for the HA group with the HA off. Mean and standard deviation for peak CAEP values for the NH group and the HA group (aided and unaided) are shown in Table 2.

The effects of intensity (40, 60, 80 dB) on CAEP responses were analyzed within the NH group using RM ANOVAs or non-parametric tests, as appropriate; complete results are shown in Table 3. For CZ amplitude, significant effects for intensity were observed at N1 (80 > 60 or 40 dB), P2 (80 > 60 or 40 dB), and N1-P2 (80 > 60 > 40 dB). For CZ latency, a significant effect of intensity was observed only at N1 (40 > 60 or 80 dB). For GFP amplitude, significant effects of intensity were observed at P1 (80 > 40 dB), N1 (80 > 60 > 40 dB), and P2 (80 > 60 > 40 dB). For GFP latency, significant effects of intensity were observed at P1 (40 > 80) and N1 (40 > 60, 80).

The effects of intensity (40, 60, 80 dB) and HA (on or off) on CAEP responses were analyzed within the HA group (aided and unaided) using RM ANOVAs or non-parametric tests, as appropriate; complete results are shown in Table 4. For CZ amplitude, significant effects for intensity were observed at P1 (80 > 60 or 40 dB), N1 (80 > 60 or 40 dB), P2 (80 > 60 or 40 dB), and N1-P2 (80, 60 > 40 dB); there was no effect of HA. For CZ latency, significant effects for intensity were observed at P1 (40 > 80) and at N1 with the HA on (40 > 60, 80). For GFP amplitude, significant effects of intensity were observed at P1 (80 > 60 > 40), N1 (80 > 60 > 40), and P2 (80 > 40); significant effects of HA were observed at P1 (on > off) and N1 (off > on). For GFP latency, significant effects of intensity were observed at P1 (40 > 60, 80) and N1 40 > 60 > 80); there was no significant effect of HA.

CAEP responses were compared between the NH group and the HA group with the HA on or off at each intensity using Kruskal–Wallis ANOVAs on ranked data; complete results are shown in Table 5. At Cz, N1 amplitude was significantly lower for the NH group than for the HA group with the HA off at all intensities. N1-P2 amplitude was significantly higher for the NH group compared to the HA group with the HA off at all intensities and with the HA on at 40 dBA. P1 latency was significantly shorter for the NH group compared to the HA group with the HA on at all intensities and with the HA off at 60 dBA. N1 latency was significantly shorter for the NH group compared to the HA group with the HA on at 40 and 60 dBA and with the HA off at 60 dBA. For GFP, N1 amplitude was significantly larger for the NH group than for the HA group with the HA off at 40 dB; there were no other significant differences between the NH group and the HA group with the HA on or off. P1 latency was significantly shorter for the NH group compared the HA group with the HA on at all intensities and with the HA off at 80 dBA. N1 latency was significantly shorter for the NH group compared the HA group with the HA on or off at 40 and 60 dBA.

### 3.3. Pupillometry

Figure 4 shows an example of pupil dilation over time at the three intensities for one NH participant. There was little change in pupil diameter at 40 dB. At 60 dB, there was greater variation in pupil dilation. The largest pupil dilation was observed at 80 dB.

Figure 5 shows boxplots of peak pupil diameter and latency at the three intensities for the NH group and the HA group (aided or unaided); mean values are shown in Table 6. In general, pupil diameter and latency increased with intensity. Within the NH group, an RM ANOVA showed significant effects of intensity on pupil diameter [F (2, 32) = 27.1, *p* < 0.001] and latency [F (2, 32) = 21.5, *p* < 0.001]. Post hoc Bonferroni pairwise comparisons showed that both pupil diameter and latency were significantly lower at 40 dB than at 60 or 80 dB (*p* < 0.001 in all cases). Within the HA group, an RM ANOVA showed a significant effect of intensity on pupil diameter [F (2, 28) = 3.8, *p* = 0.036]; post hoc Bonferroni pairwise comparisons showed that pupil diameter was significantly lower at 40 dB than at 80 dB (*p* = 0.032). There was no significant effect of HA (on or off) on pupil diameter or latency. A Kruskal–Wallis ANOVA on ranked data showed a difference across the NH and HA groups only for pupil latency at 80 dBA (dF = 2, H = 9.5, *p* = 0.009). After Bonferroni adjustment for multiple comparisons, latency was significantly higher for the NH group than for the HA group with the HA off (*p* = 0.008).

### 3.4. Relationships among Behavioral and Objective Measures Using FCPA

FCPA was performed on the subjective loudness, CAEP, and pupillometry data to explore the relationships between loudness perception and explanatory objective measures, as well as the relationships among the objective measures. The right panels of Figure 6 show the correlation matrix among all variables. The right panels of Figure 6 visualize these correlations. The strength of the correlation between the dependent variable of loudness (the center of the circle) and the explanatory objective measures are represented by the concentric circles; r values ≥ 0.5 were considered to be the strongest explanatory variables, and data within the red circle indicate significant relationships (*p* < 0.05). The closer the variable is to the center of the plot, the stronger the correlation. The distance among the explanatory variables indicates the degree of inter-correlation. When points are close together, the variables are strongly and positively correlated. When points are diametrically opposed, the variables are strongly and negatively correlated. When points are equidistant from the origin, there is no significant inter-correlation [69,70].

For the NH group (top panels of Figure 6), loudness perception was positively correlated with [N1-P2] amplitude at Cz (r = 0.64), pupil diameter (r = 0.59), and pupil latency (r = 0.61), and negatively correlated with N1 amplitude GFP (r = −0.57). Pupil diameter and latency were in close proximity, indicating substantial inter-correlation (r = 0.59). Pupil latency and N1 latency GFP were diametrically opposed, indicating some inter-correlation (r = −0.37). The remaining objective measures showed relatively weak relationships to loudness perception (r < 0.05).

With the HA on, loudness perception for the HA group was positively correlated with P1 amplitude GFP (r = 0.56), and negatively correlated with P1 latency at Cz (r = −0.6), N1 latency at Cz (r = −0.6), and N1 latency GFP (r = −0.68). N1 latency at Cz and N1 latency GFP latency were in close proximity, indicating substantial inter-correlation (r = 0.82). With the HA off, loudness perception was positively correlated with P1 amplitude GFP (r = 0.51) and N1 amplitude GFP (r = 0.6).

## 4. Discussion

To the best of our knowledge, this is the first exploratory study to compare loudness, pupillometry, and CAEPs in NH and HA listeners. Most previous studies have compared such subjective and objective measures in cochlear implant (CI) rather than HA users [20,21]. The goal of this study was to determine if pupillometry and/or CAEP could be a marker of auditory listening comfort. Results showed a strong impact of intensity level on loudness ratings, pupillometry, and CAEP responses in the NH and HA groups (with the HA on or off). For the HA group, there was little difference in behavioral responses when the HA was on or off. Significant relationships were observed between loudness ratings and some CAEP and pupillometry measures, though these differed between the NH and HA groups. Below we discuss the findings in greater detail.

### 4.1. Effect of Intensity Level on Loudness Perception, CAEP, and Pupillometry

Not surprisingly, loudness perception was closely related to intensity level for the NH and HA groups (Figure 2). No significant difference was observed between the NH group and the HA group with the HA on or off. Within the NH and HA groups, loudness ratings significantly increased with intensity. Within the HA group, loudness ratings were significantly higher with the HA on than off at 40 and 60 dB, but not at 80 dB. This was likely due to the compression in the HA, where softer sounds would be amplified, and loud sounds would be peak-limited.

For all groups, stimulus intensity affected CAEP waveform morphology (Figure 3). For the NH group, significant effects of intensity were observed for all CAEP responses except for P1 amplitude at Cz, P2 latency at Cz, and P2 latency GFP (Table 3). Similarly, significant effects for intensity were observed for all CAEP responses, except for P2 latency at Cz, and P2 latency GFP (Table 4). Shorter peak latencies signify decreased neural conduction time, and higher amplitudes represent increased response strength [40,41,42,43,44,45,46,47,48,49,50,51,52,53]. Similar patterns were observed in the present study, especially for P1 and N1 responses. Neural encoding for sound intensity has been directly linked to N1 peak amplitude [49,58].

Some significant differences in CAEP responses were observed between the NH group and the HA group, mostly for P1 latency and N1 amplitude and latency (Table 5). With the HA off, peak amplitude negativity was generally smaller for the HA group than for the NH group. This may have corresponded to the lower loudness ratings at 40 and 60 dB when the HA was off, compared to the NH group. With the HA on or off, peak latency was generally longer for the HA group than for the NH group. When the HA was on, compressor time constants may have affected latency. When the HA was off, poorer audibility may have resulted in longer latency [71].

Significant effects of intensity on pupillometry were observed for the NH and HA groups (Figure 5). For the NH group, peak pupil diameter and latency significantly increased across all intensities. These results agree with previous NH studies that showed larger pupil responses with increasing intensity [46,47,48]. The increase in pupil response could be explained by the nature of the stimuli, which were sudden and novel from trial to trial, and may have evoked automatic attentional effects. As hearing is important for warning sounds, high intensity levels could be interpreted as an alarm signal or an environmental stressor, which would disturb the autonomic nervous system. As pupil dilation depends on the autonomic nervous system [72], it adapts to the environmental demands, including auditory stimulation. Kahneman described pupillometry as an index of “load on attention capacity” [73]; high intensity stimulation likely leads to greater attention. Thus, the increase in pupil dilation with intensity could be interpreted as an automatic direction of attention to the stimulus. Pupillometry can also provide an estimate of mental effort and has been correlated with activity in the locus coeruleus [74]. High intensity may require more time to arrive at peak dilation because of noradrenaline discharge [60], resulting in longer peak latency, as observed in the present NH group. 

For the HA group, peak dilation was significantly different only between 40 and 80 dB, and only when the HA was on. There was no significant difference in peak latency across intensities with the HA on or off. When the HA was on, peak intensity may have been compressed and lower intensities would be likely be amplified, resulting in less differentiation across intensity than would occur with NH listeners. When the HA was off, there may have been a reduction in audibility that may have limited pupil response. 

Interestingly, the only significant difference observed between the NH group and HA groups was for pupil dilation at 80 dB when the HA was off. While pupil response to intensity was more pronounced for the NH group, the range of pupil responses was generally similar between the NH group and the HA group when the HA was on or off (Table 6).

### 4.2. Effect of HA Amplification on Loudness Perception, CAEP, and Pupillometry Responses

Because intensity significantly affected loudness ratings and CAEP responses in the HA group, and because lower sounds would be amplified when the HA was on, we expected that HA amplification would significantly affect loudness perception and CAEP response. HA amplification significantly increased loudness ratings only at 40 and 60 dB, with no difference at 80 dB. Note that the compression ratio was greater for high and moderate sound levels (1.7 ± 0.7) than for low sound levels (1.3 ± 0.4), which may have contributed to this finding. Alternatively, there may have been some physiological saturation at the higher 80 dB intensity level that was unaffected by HA amplification [75]. HA amplification significantly increased CAEP responses for P1 and N1 amplitude GFP across all intensities, for P2 amplitude GFP at 80 dB (Table 4). No significant differences for HA amplification were observed for latency measures. The greater neural recruitment associated with HA amplification may increase CAEP amplitude in some cases but may not affect auditory processing time at the cortical level [76].

Previous studies in which NH listeners were tested with or without simulations of HA amplification showed that CAEP amplitudes decreased with HA amplification [63] or were unaffected by HA amplification [62,64]. Other studies in individuals with hearing loss showed no change in cortical responses with HA amplification [77]. Karawani et al. (2018) found increased N1 and P2 peak amplitude with HA amplification that was correlated with improvements in working memory [78], suggesting that HA experience may enhance cortical sound processing and improve cognitive function. In the present study, we did not observe a robust effect of HA amplification, even though participants had used their HA for at least one year. One possible explanation is that daily auditory stimulation via HA did not alter the neurophysiological representation of sound at the level of the auditory cortex.

The mean changes in loudness across intensities were different with and without HA amplification. With the HA off, the mean change in loudness was 2.5 from 40 to 60 dB, and 2.9 from 60 to 80 dB; with the HA on, the mean change in loudness was 2.1 from 40 to 60 dB, and 1.7 from 60 to 80 dB. As such, changes in loudness with intensity were much smaller with HA amplification than with changes in intensity. It is also possible that HA amplification changes more than just the intensity of a stimulus. HA amplification may differently affect neural responses to changes in intensity, although this may be truer for more complex stimuli (e.g., speech, noise) than for the pure tone stimuli used in this study. Taken together, the present data suggest that cortical responses and pupillometry may not be good approaches to evaluate the effects of HA amplification.

### 4.3. Relationships between Subjective and Objective Measures of Auditory Intensity

FCPA revealed few significant relationships between behavioral and objective responses to intensity; the observed relationships differed among the NH groups and the HA group with the HA on or off. For the NH group, loudness was significantly associated with [N1-P2] amplitude at Cz, N1 amplitude GFP, as well as pupil dilation and latency. For the HA group, loudness was significantly associated with P1 and N1 latency at Cz, P1 amplitude GFP, and N1 latency GFP; with the HA off, loudness was significantly associated with P1 and N1 amplitude GFP.

Loudness was significantly associated with pupil dilation and latency only in the NH group. The mean dynamic range for pupil dilation was 0.13 mm for the NH group, 11 mm for the HA group with the HA on, and 7 mm for the HA group with the HA off (Table 6). Increasing age has been associated with reduced dynamic range of pupil dilation in older than in younger listeners [79]. Age might differentially affect different components of the pupil response reflecting parasympathetic versus sympathetic effects [80]. In this study, the mean age of the NH group (64.0 ± 3.9 years) was 12 years younger than that of the HA group (76.0 ± 7.0 years). The older age of the HA group may partly explain the lack of association between loudness and pupil response, due to the reduced dynamic range of pupil dilation.

Concerning CAEPs, Bakhos et al. (2014) showed that some pediatric HA users exhibit abnormal temporal brain function (absence of N1c) that may underlie language impairment [81]. Of course, children react differently than adults, and their hearing loss may have other consequences for children (e.g., language development) than for elderly adults (e.g., CAEPs or pupil responses). Behavioral and objective responses to sound intensity should also be measured in children. Indeed, the principal interest in using objective measures is responses to sound intensity where behavioral measures are difficult to measure, as is the case for young children.

### 4.4. Limits to Study

While this study demonstrated the impact of intensity on CAEPs and pupillometry, the number of participants was limited. Indeed, a power analysis with G *power software indicated that for power = 0.95, 43 participants would be needed for each group. Additional studies with a larger cohort should be conducted to confirm the present findings. In addition, only 1-kHz tone bursts were used, and it remains unclear how tone frequency might affect responses to sound intensity. Indeed, loudness has been shown to depend on spectral [82,83,84] and temporal [85,86,87] properties of sound, as well as other factors.

Age at testing was significantly different between both the NH and HA groups and might have contributed to group differences by affecting neural responses. However, it is difficult to recruit adults up to 70 years old that still have normal hearing thresholds due to presbycusis. Moreover, HA devices are very complex. In this study, only Oticon and Bernafon in-ear HAs were used. HA signal processing, such as channel-specific compression time constants, noise reduction algorithms, and adaptive directionality, may affect CAEPs [88]. It may be preferable to conduct this study using the exact same devices and settings for all HA participants.

Finally, good HA outcomes likely cannot be reduced to auditory comfort markers such as CAEP or pupillometry. Many factors can contribute to good HA outcomes, such as device-specific (e.g., directional microphones, signal processing, gain settings) and patient-specific variables (age, attention, motivation, biology, personality, lifestyle) [63].

## 5. Summary of the Results

SNHL can distort perception of sound intensity. In this study, behavioral (loudness) and objective responses (CAEPs, pupillometry) to sound intensity were measured in NH and HA participants (with the HA on or off). For all groups, loudness increased with intensity; for the HA group, loudness was significantly higher with the HA on than off only at the low-to-mid intensities. Most CAEPs showed a significant response to intensity in the NH and HA groups; there was little effect of HA amplification on CAEPs. Similarly, there was a significant response in pupil dilation and latency to intensity in the NH and HA groups; there was little effect of HA amplification on pupil response. FPCA showed only a few significant relationships between behavioral and objective measures. Further research is needed to better understand the relationship between stimulus intensity, loudness perception, and objective measures.

## Figures and Tables

**Figure 1 brainsci-12-00392-f001:**
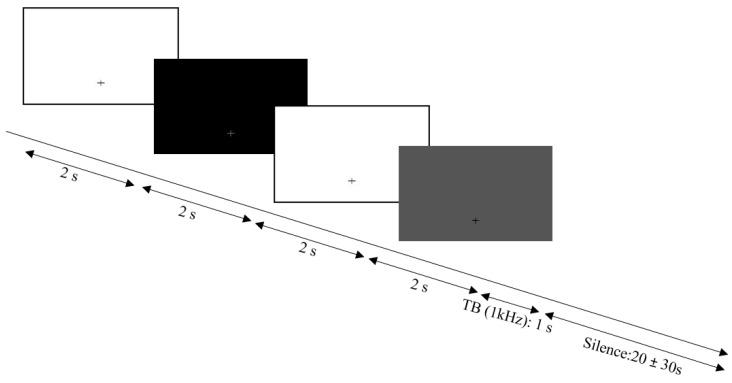
Pupillometry paradigm. Progression of the visual stimuli: first a white image was presented for 2 s, followed by a black image for 2 s, followed by a white image for 2 s, followed finally by a gray image, which remained present during the experimental run. A 1-kHz tone burst was presented at an interval of 20–30 s.

**Figure 2 brainsci-12-00392-f002:**
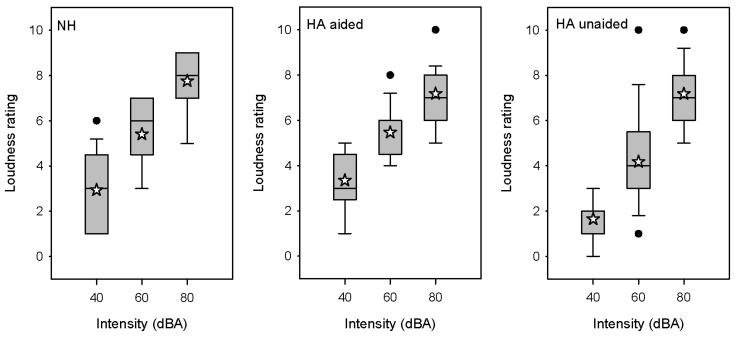
Boxplots of loudness ratings for 1-kHz tone bursts at 40, 60, or 80 dBA for the NH group, the HA group (**aided**), and the HA group (**unaided**). The boxes show the 25th and 75th percentiles, the error bars show the 10th and 90th percentiles, the filled circles show outliers, the horizontal lines show the median, and the white stars show the mean.

**Figure 3 brainsci-12-00392-f003:**
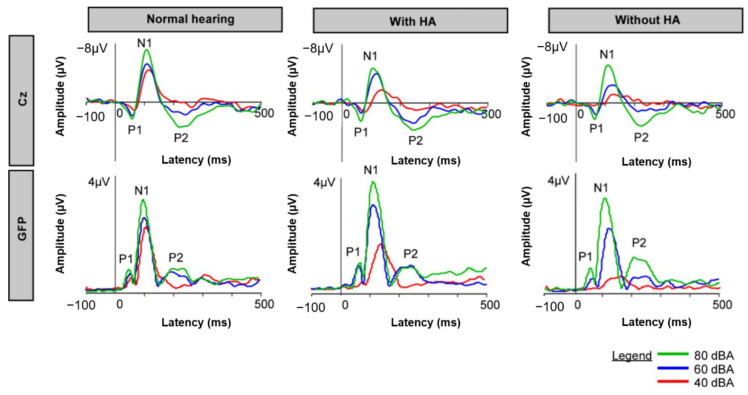
Mean CAEP responses recorded at Cz (**top**) or for GFP (**bottom**) with a 1-kHz tone burst presented at 40, 60, or 80 dBA for the NH group and the HA group (aided and unaided).

**Figure 4 brainsci-12-00392-f004:**
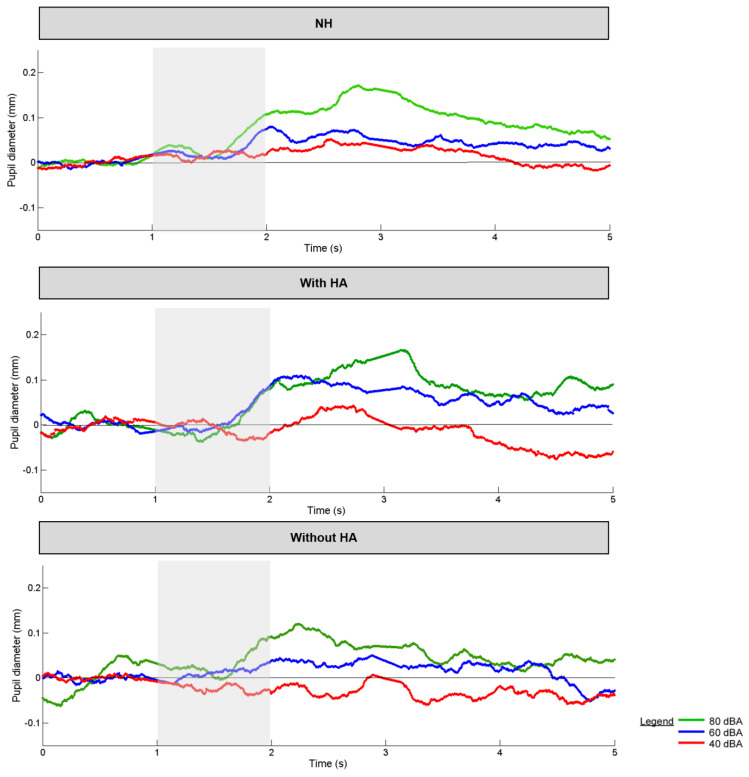
Peak pupil diameter as a function of time for mean NH group and the HA group (**aided and unaided**) after a tone burst at 1 kHz with 1 s length (**grey shaded**).

**Figure 5 brainsci-12-00392-f005:**
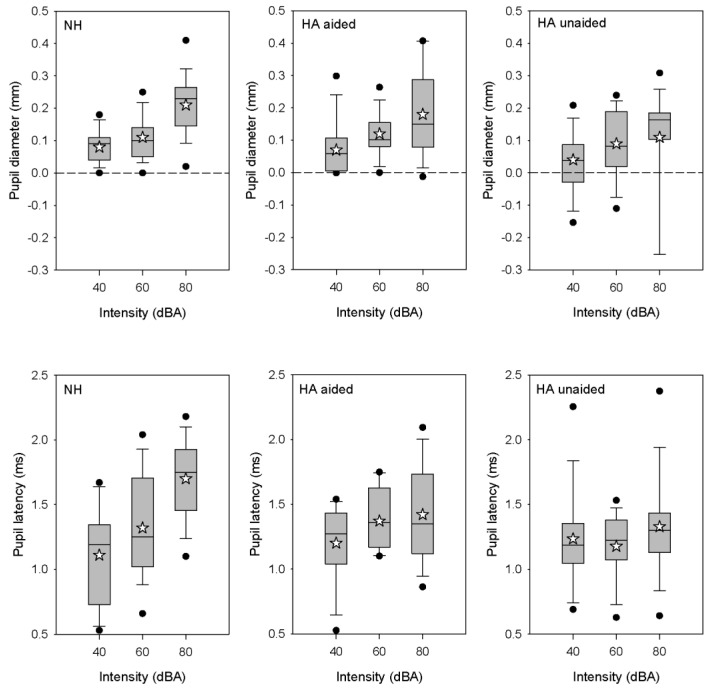
Boxplots of peak pupil diameter (**top**) and latency (**bottom**) in response to 1-kHz tone bursts at 40, 60, or 80 dBA for the NH for the NH group and the HA group with the HA on or off. The boxes show the 25th and 75th percentiles, the error bars show the 10th and 90th percentiles, the filled circles show outliers, the horizontal lines show the median, and the white stars show the mean.

**Figure 6 brainsci-12-00392-f006:**
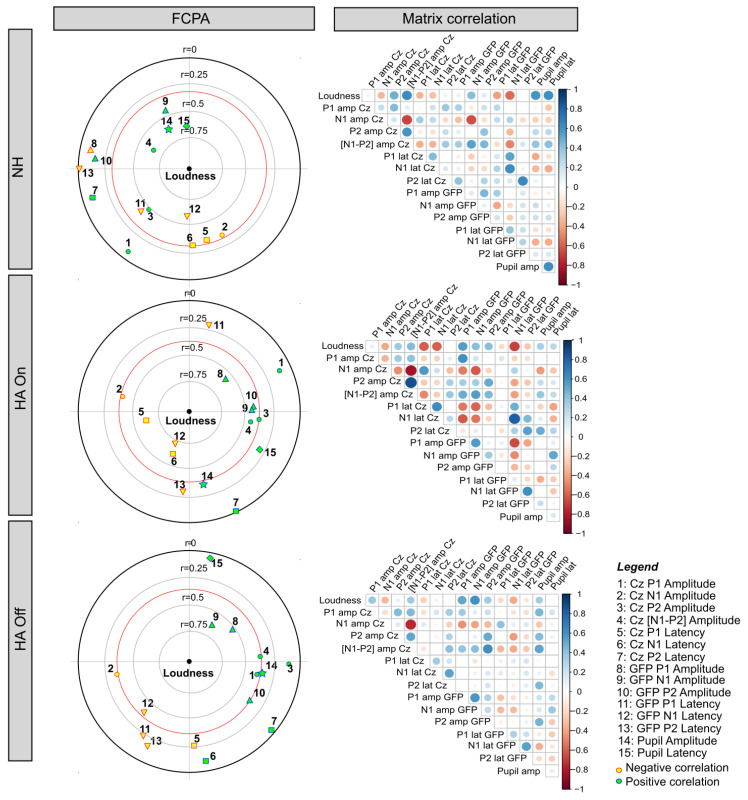
Left panels: FPCA plots of between loudness perception and objective responses to stimuli for the NH and HA groups (HA on or off); the center represents the variable to explain (loudness) and explanatory variables (pupillometry and CAEPs). The circles represent Cz amplitude, the squares represent Cz latency, the up triangles represent GFP amplitude, the down triangles represent GFP latency, the stars represent pupil diameter, and the diamonds represent pupil latency. The green and yellow symbols represent positive and negative correlations between the dependent and explanatory variables, respectively. The explanatory variables within the red circle were significantly correlated with the dependent variable (*p* < 0.05). Right panels: Correlation matrices among the behavioral and objective measures. The bar to the right of the matrices shows the color coding for the correlation coefficients, ranging from −1 to 1. Within the matrices, the color and size of the circle represents the strength of correlation between two variables.

**Table 1 brainsci-12-00392-t001:** Demographic information for the HA group.

		Age	Unaided PTA (dB HL)		Tone HA Gain	Unaided SAT (dB SPL)		Speech HA Gain
Participant	Sex	(yrs.)	Left	Right	(dB)	Left	Right	(dB)
HA-1	M	62	26.3	27.5	5.0	50.0	55.0	7.0
HA-2	F	75	53.8	50.0	16.3	65.0	40.0	16.0
HA-3	F	76	42.5	57.5	8.8	60.0	48.0	3.0
HA-4	F	74	21.3	27.5	13.8	61.0	50.0	16.0
HA-5	M	76	31.3	42.5	8.8	47.0	50.0	5.0
HA-6	F	65	70.0	61.3	18.8	38.0	38.0	4.0
HA-7	F	80	55.0	52.5	21.3	35.0	54.0	20.0
HA-8	M	81	57.5	65.0	33.8	62.0	57.0	28.0
HA-9	M	88	51.3	50.0	28.8	80.0	70.0	19.0
HA-10	M	72	70.0	62.5	11.3	40.0	52.0	6.0
HA-11	M	81	56.3	61.3	31.3	67.0	67.0	25.0
HA-12	F	83	48.8	41.3	11.3	55.0	50.0	7.0
HA-13	M	82	52.5	48.8	20.0	60.0	55.0	21.0
HA-14	F	76	46.3	45.0	18.8	52.0	55.0	14.0
HA-15	M	66	36.3	37.5	16.3	37.0	47.0	6.0
HA-16	F	73	40.0	36.3	8.8	40.0	40.0	9.0
HA-17	F	83	48.8	41.3	6.3	50.0	50.0	6.0
Mean	8 M, 9 F	76.1	47.7	47.5	16.4	52.9	51.7	12.5
SD		7.0	14.0	11.7	8.6	12.5	8.5	8.0

M = male; F = female; PTA = pure-tone average thresholds across 0.5, 1.0, 2.0, and 4.0 kHz; SAT = speech audibility threshold; SD = standard deviation.

**Table 2 brainsci-12-00392-t002:** Mean ± standard deviation for peak CAEP amplitude and latency at Cz electrode and for global field power (GFP), for the NH group and the HA group (aided or unaided) at each intensity.

Intensity	Peak			NH	HA Aided	HA Unaided
40 dBA	P1	Amplitude (µV)	Cz	1.4 ± 1.2	1.5 ± 1.3	0.9 ± 1.2
			GFP	0.9 ± 0.5	0.8 ± 0.3	0.7 ± 0.2
		Latency (ms)	Cz	64.8 ± 10.7	78.4 ± 19.4	77.7 ± 23.3
			GFP	58.8 ± 14.1	76.1 ± 12.2	71.9 ± 20.7
	N1	Amplitude (µV)	Cz	−4.9 ± 2	−2.1 ± 2.4	−1.8 ± 2.7
			GFP	2.4 ± 0.7	2.0 ± 0.9	0.8 ± 0.3
		Latency (ms)	Cz	116.8 ± 10.3	138.9 ± 29.9	128.7 ± 29.8
			GFP	114.8 ± 15.3	136.1 ± 18.5	129.9 ± 16.4
	P2	Amplitude (µV)	Cz	0.7 ± 1.5	1.0 ± 1.7	0.8 ± 1.2
			GFP	1.0 ± 0.4	1.1 ± 0.4	0.9 ± 0.4
		Latency (ms)	Cz	183.5 ± 30.7	192.2 ± 42.9	181.2 ± 40.2
			GFP	188.3 ± 19.9	208.2 ± 28	204.2 ± 24.9
	N1-P2	Amplitude (µV)	Cz	5.6 ± 1.8	3.2 ± 2.6	3.0 ± 2
60 dBA	P1	Amplitude (µV)	Cz	2.1 ± 1.7	1.6 ± 1.8	1.4 ± 1.9
			GFP	1.0 ± 0.4	1.2 ± 0.4	1.1 ± 0.3
		Latency (ms)	Cz	58.7 ± 8	69.9 ± 8.6	72.9 ± 11.6
			GFP	53.0 ± 8.6	63.1 ± 8.2	62.6 ± 15.9
	N1	Amplitude (µV)	Cz	−5.6 ± 2.7	−4.5 ± 2.1	−3.0 ± 2.9
			GFP	2.9 ± 0.9	3.3 ± 0.9	2.7 ± 1.1
		Latency (ms)	Cz	110.4 ± 7.7	120.4 ± 10.3	132 ± 20.7
			GFP	102 ± 6.9	113.9 ± 11.7	120.1 ± 15.1
	P2	Amplitude (µV)	Cz	2.5 ± 1.9	2.6 ± 2.6	1.8 ± 1.6
			GFP	1.4 ± 0.6	1.3 ± 0.5	1.3 ± 0.5
		Latency (ms)	Cz	199.7 ± 40.7	194.8 ± 29.2	201.8 ± 30.8
			GFP	191.6 ± 25.7	198.1 ± 25.8	199.9 ± 22
	N1-P2	Amplitude (µV)	Cz	8.1 ± 2.4	7.1 ± 2.9	5.0 ± 2.2
80 dBA	P1	Amplitude (µV)	Cz	2.6 ± 2.6	3.1 ± 2.4	2.7 ± 1.1
			GFP	1.2 ± 0.4	1.5 ± 0.5	1.3 ± 0.5
		Latency (ms)	Cz	56.9 ± 5.4	62.4 ± 9.1	64.5 ± 15.9
			GFP	51.4 ± 6.4	61.1 ± 6.9	58.5 ± 8.2
	N1	Amplitude (µV)	Cz	−7.8 ± 2.7	−5.5 ± 3	−5.8 ± 3.3
			GFP	3.5 ± 1.2	4.2 ± 1.1	3.7 ± 1.5
		Latency (ms)	Cz	107.1 ± 7.4	114.5 ± 12.8	116 ± 15.9
			GFP	101.9 ± 5.6	107.8 ± 10.6	109.8 ± 11.4
	P2	Amplitude (µV)	Cz	4.1 ± 2.9	2.8 ± 3.3	2.7 ± 2.9
			GFP	1.6 ± 0.9	1.4 ± 0.7	1.9 ± 1.1
		Latency (ms)	Cz	200.8 ± 31.1	184.1 ± 31.3	185.7 ± 29.9
			GFP	196.8 ± 22.3	194.9 ± 20.7	204 ± 29.8
	N1-P2	Amplitude (µV)	Cz	11.9 ± 3.3	8.4 ± 4.1	8.2 ± 4.7

GFP = global field power; NH = normal hearing; HA = hearing aid.

**Table 3 brainsci-12-00392-t003:** Results of RM ANOVAs performed on NH CAEP amplitude and latency data at Cz and for global field power (GFP); in cases where assumptions of normality and equal variance were violated, non-parametric tests were performed (shown in lower part of the table). Significant effects are indicated by asterisks and italics; post hoc significant differences are shown after Bonferroni or Tukey correction for multiple comparisons.

RM ANOVA					
Peak	Factor	dF, res	F	*p*	Post Hoc Bonferroni
N1 amp Cz	Intensity	2, 32	18.2	*<0.001 **	60, 40 > 80
P2 amp Cz	Intensity	2, 32	14.5	*<0.001 **	80, 60 > 40
N1-P2 amp Cz	Intensity	2, 32	61.9	*<0.001 **	80 > 60 > 40
N1 lat Cz	Intensity	2, 32	14.5	*<0.001 **	40 > 60, 80
P1 amp GFP	Intensity	2, 32	3.3	*0.049 **	80 > 40
N1 amp GFP	Intensity	2, 32	24.0	*<0.001 **	80 > 60 > 40
P2 lat GFP	Intensity	2, 32	1.7	0.196	
**Kruskal–Wallis ANOVA on ranked data**					
**Peak**	**Factor**	**dF**	**χ^2^**	** *p* **	**Post Hoc Tukey**
P1 amp Cz	Intensity	2	4.6	0.101	
P1 lat Cz	Intensity	2	13.6	0.001 *	80 > 40
P2 lat Cz	Intensity	2	4.4	0.113	
P2 amp GFP	Intensity	2	19.2	*<0.001 **	80 > 60, 40
P1 lat GFP	Intensity	2	8.9	*0.012 **	40 > 80
N1 lat GFP	Intensity	2	23.7	*<0.001 **	40 > 60, 80

**Table 4 brainsci-12-00392-t004:** Results of RM ANOVAs performed on HA CAEP amplitude and latency data (aided and unaided) at Cz and for global field power (GFP); in cases where assumptions of normality and equal variance were violated, non-parametric tests were performed (shown in lower part of the table). For the HA factor, the HA was on or off. Significant effects are indicated by asterisks and italics; post hoc significant differences are shown after Bonferroni or Tukey correction for multiple comparisons.

RM ANOVA					
Peak	Factor	dF, res	F	*p*	Post Hoc Bonferroni
P1 amp Cz	HA	1, 32	1.4	0.260	
	Intensity	2, 32	8.7	*<0.001 **	80 > 60, 40
	HA x intensity	2, 32	0.1	0.872	
P2 amp Cz	HA	1, 32	1.3	0.275	
	Intensity	2, 32	5.8	*0.007*	80, 60 > 40
	HA x intensity	2, 32	0.5	0.619	
P1 lat Cz	HA	1, 32	0.1	0.818	
	Intensity	2, 32	11.0	*<0.001 **	40 > 80
	HA x intensity	2, 32	0.2	0.804	
P2 lat Cz	HA	1, 32	<0.1	0.857	
	Intensity	2, 32	1.4	0.260	
	HA x intensity	2, 32	0.8	0.458	
P1 amp GFP	HA	1, 32	5.0	*0.039 **	HA on > HA off
	Intensity	2, 32	27.4	*<0.001 **	80 > 60 > 40
	HA x intensity	2, 32	0.3	0.769	
N1 amp GFP	HA	1, 32	32.1	*<0.001 **	HA-a > HA-un
	Intensity	2, 32	56.6	*<0.001 **	80 > 60 > 40
	HA x intensity	2, 32	3.0	0.063	
P2 amp GFP	HA	1, 32	0.8	0.391	
	Intensity	2, 32	10.8	*<0.001 **	80 > 40
	HA x intensity	2, 32	5.9	*0.007 **	HA on: 80 > 60 > 40; 80: HA on > HA off
P1 lat GFP	HA	1, 32	2.0	0.179	
	Intensity	2, 32	10.7	*<0.001 **	40 > 60, 80
	HA x intensity	2, 32	0.2	0.800	
N1 lat GFP	HA	1, 32	0.1	0.783	
	Intensity	2, 32	31.0	*<0.001 **	40 > 60 > 80
	HA x intensity	2, 32	1.9	0.163	
P2 lat GFP	HA	1, 32	0.3	0.583	
	Intensity	2, 32	0.9	0.432	
	HA x intensity	2, 32	0.8	0.452	
**Kruskal–Wallis ANOVA on ranked data**					
**Peak**	**Factor**	**dF**	**χ^2^**	** *P* **	**Post Hoc Tukey**
N1 amp Cz	HA, intensity	5	51.2	*<0.001 **	60, 40 > 80
N1-P2 amp Cz	HA, intensity	5	44.9	*<0.001 **	80, 60 > 40
N1 lat Cz	HA, intensity	5	21.7	*<0.001 **	HA on: 40 > 60, 80

**Table 5 brainsci-12-00392-t005:** Results of Kruskal–Wallis ANOVAs on ranked data. CAEP amplitude and latency data (aided and unaided) were compared across the NH and HA groups (aided or unaided) at Cz and for global field power (GFP) at each intensity. Significant effects are indicated by asterisks and italics; post hoc significant differences are shown after Bonferroni correction (adjusted *p* = 0.025) for multiple comparisons (Dunn) where NH data were compared to HA data with the HA on or off.

CAEP	Intensity	dF	H	*p*	Post Hoc Dunn’s
P1 amp Cz	40	2	1.7	0.423	
	60	2	1.2	0.554	
	80	2	1.6	0.557	
N1 amp Cz	40	2	19	*<0.001 **	NH < HA off
	60	2	8.1	*0.018 **	NH < HA off
	80	2	7.1	*0.029 **	NH < HA off
P2 amp Cz	40	2	1.1	0.569	
	60	2	1.4	0.504	
	80	2	1.7	0.437	
N1-P2 amp Cz	40	2	14.1	*<0.001 **	NH > Ha off, HA on
	60	2	12.4	*0.002 **	NH > HA off
	80	2	7.3	*0.027 **	NH > HA off
P1 lat Cz	40	2	6.5	*0.039 **	NH < HA on
	60	2	17	*<0.001 **	NH < HA off, HA on
	80	2	7.5	*0.024 **	NH < HA on
N1 lat Cz	40	2	8.5	*0.014 **	NH < HA on
	60	2	16.2	*<0.001 **	NH < HA off, HA on
	80	2	4.1	0.132	
P2 lat Cz	40	2	1.1	0.570	
	60	2	0.5	0.779	
	80	2	3	0.223	
P1 amp GFP	40	2	1.1	0.586	
	60	2	2.2	0.328	
	80	2	4	0.135	
N1 amp GFP	40	2	30.3	*<0.001 **	NH > HA off
	60	2	4.8	0.093	
	80	2	3	0.218	
P2 amp GFP	40	2	2.8	0.243	
	60	2	0.4	0.803	
	80	2	1	0.592	
P1 lat GFP	40	2	10.9	*0.004 **	NH < HA on
	60	2	7.8	*0.020 **	NH < HA on
	80	2	13.9	*<0.001 **	NH < HA on, HA off
N1 lat GFP	40	2	13.2	*0.001 **	NH < HA on, HA off
	60	2	14.7	*<0.001 **	NH < HA on, HA off
	80	2	3.5	0.170	
P2 lat GFP	40	2	4.2	0.123	
	60	2	0.5	0.760	
	80	2	0.9	0.645	

**Table 6 brainsci-12-00392-t006:** Mean ± standard deviation for peak pupil diameter and latency for the NH group and the HA group (aided or unaided) at each intensity.

	Intensity	NH	HA Aided	HA Unaided
Peak dilation (mm)	40 dBA	0.08 ± 0.05	0.07 ± 0.08	0.04 ± 0.09
	60 dBA	0.11 ± 0.07	0.07 ± 0.08	0.09 ± 0.10
	80 dBA	0.21 ± 0.09	0.18 ± 0.13	0.11 ± 0.18
Peak latency (ms)	40 dBA	1.11 ± 0.36	1.20 ± 0.29	1.23 ± 0.36
	60 dBA	1.32 ± 0.38	1.37 ± 0.23	1.18 ± 0.25
	80 dBA	1.70 ± 0.31	1.42 ± 0.37	1.33 ± 0.38

## Data Availability

Data are available in the Appendix A.

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
