# Peer review of "Relationship between Behavioral and Objective Measures of Sound Intensity in Normal-Hearing Listeners and Hearing-Aid Users: A Pilot Study"

_brainsci, 2022, doi:10.3390/brainsci12030392_

Round 1

Reviewer 1 Report

General comments:

The manuscript is related to searching for some objective methods to evaluate auditory comfort. This problem is important and when solved could lead to finding objective, possibly automatic,  methods of hearing aid fitting. This is an interesting and well written manuscript. I believe that finding objective equivalent to psychoacoustic methods is the most important topic area in audiology.

My main reservation to this study is small study group. I believe that it should be at least 40-50 subjects in each group, but not less than 20. The authors mention this in the limitation section but this is not enough. It could be a limitation when there would be 20-25 subjects but less than 20 is not acceptable in my opinion. The only way I see this manuscript accepted if in the title it would be added that it is a pilot/preliminary study. However I strongly encourage the authors to expand the number of subjects to 40-50. It would strengthen the study.

Specific comments:

Introduction:

The introduction only describes some objective methods that are of interest of the authors. However there is lack of some stronger motivation. The rationale for the study must be more clearly stated. What we do know about this problem objective assessment of auditory comfort? Are there any studies on that topic? What do we know, what we do not know? How the authors want to surpass or extend the current knowledge?

Consider discussing results of these studies:

Bianchi F, Wendt D, Wassard C, Maas P, Lunner T, Rosenbom T, Holmberg M. Benefit of Higher Maximum Force Output on Listening Effort in Bone-Anchored Hearing System Users: A Pupillometry Study. Ear Hear. 2019 Sep/Oct;40(5):1220-1232. doi: 10.1097/AUD.0000000000000699.

Wagner AE, Nagels L, Toffanin P, Opie JM, Başkent D. Individual Variations in Effort: Assessing Pupillometry for the Hearing Impaired. Trends Hear. 2019 Jan-Dec;23:2331216519845596. doi: 10.1177/2331216519845596.

Seifi Ala T, Graversen C, Wendt D, Alickovic E, Whitmer WM, Lunner T. An exploratory Study of EEG Alpha Oscillation and Pupil Dilation in Hearing-Aid Users During Effortful listening to Continuous Speech. PLoS One. 2020 Jul 10;15(7):e0235782. doi: 10.1371/journal.pone.0235782.

Micula A, Rönnberg J, Fiedler L, Wendt D, Jørgensen MC, Larsen DK, Ng EHN. The Effects of Task Difficulty Predictability and Noise Reduction on Recall Performance and Pupil Dilation Responses. Ear Hear. 2021 Nov-Dec 01;42(6):1668-1679. doi: 10.1097/AUD.0000000000001053.

Fiedler L, Seifi Ala T, Graversen C, Alickovic E, Lunner T, Wendt D. Hearing Aid Noise Reduction Lowers the Sustained Listening Effort During Continuous Speech in Noise-A Combined Pupillometry and EEG Study. Ear Hear. 2021 Nov-Dec 01;42(6):1590-1601. doi: 10.1097/AUD.0000000000001050.

Methods:

The study group is too small. It should be minimum of 20 subjects in each subgroup (It would be the best to have 40-50 subjects). For some analyses only 15 subjects were taken. The authors should at least perform power analysis indicating what number of the subjects would be required for establishing significance.

Please describe briefly the whole procedure. Which tests where done first, which next? Were all tests made at the same session? How long took all the tests? Did the subjects got tired after all the testing? Was the problem of falling them asleep? How the authors tried to eliminate this factor?

In which program other analyses than FCPA were done? Also in R?

Results:

I suspect that for correlation analysis less than 20 subjects is way too small. I am afraid that some of the results might be incidental.

All figures are very well prepared however the figure 6 has too small markers and captions.

Discussion:

The first sentences of the discussion might be also the basis of providing better rationale in the Introduction.

“4.1. Effect of intensity level on loudness perception, CAEP and pupillometry” – there seem to be no need to mention Bonferroni correction in

Conclusions:

Please consider rewriting or changing the title of this section to “Summary of the results” as there are no real conclusions here except that “Further research is needed to…” which is only an obvious general statement.

Editing errors:

Too large font for section title “2.3.1. Categorical loudness scaling”.

Hard to distinguish subsection titles for “Cortical auditory evoked potentials (CAEPs)”: “Stimuli” and “Electroencephalogram (EEG) data recording”

Pages 6-7 – different font sizes throughout paragraphs.

“4.4. Limits to study” – change “unlear” to “unclear”

Author Response

General comments:

The manuscript is related to searching for some objective methods to evaluate auditory comfort. This problem is important and when solved could lead to finding objective, possibly automatic, methods of hearing aid fitting. This is an interesting and well written manuscript. I believe that finding objective equivalent to psychoacoustic methods is the most important topic area in audiology.

My main reservation to this study is small study group. I believe that it should be at least 40-50 subjects in each group, but not less than 20. The authors mention this in the limitation section but this is not enough. It could be a limitation when there would be 20-25 subjects but less than 20 is not acceptable in my opinion. The only way I see this manuscript accepted if in the title it would be added that it is a pilot/preliminary study. However I strongly encourage the authors to expand the number of subjects to 40-50. It would strengthen the study.

>> While we agree that it would be great to have a greater number of subjects, this is not feasible at this time. As you suggested, we have retitled as: “Relationship between behavioral and objective measures of sound intensity in normal-hearing listeners and hearing-aid users: a pilot study.”

Specific comments:

Introduction:

The introduction only describes some objective methods that are of interest of the authors. However there is lack of some stronger motivation. The rationale for the study must be more clearly stated. What we do know about this problem objective assessment of auditory comfort? Are there any studies on that topic? What do we know, what we do not know? How the authors want to surpass or extend the current knowledge?

Consider discussing results of these studies:

Bianchi F, Wendt D, Wassard C, Maas P, Lunner T, Rosenbom T, Holmberg M. Benefit of Higher Maximum Force Output on Listening Effort in Bone-Anchored Hearing System Users: A Pupillometry Study. Ear Hear. 2019 Sep/Oct;40(5):1220-1232. doi: 10.1097/AUD.0000000000000699.

Wagner AE, Nagels L, Toffanin P, Opie JM, Başkent D. Individual Variations in Effort: Assessing Pupillometry for the Hearing Impaired. Trends Hear. 2019 Jan-Dec;23:2331216519845596. doi: 10.1177/2331216519845596.

Seifi Ala T, Graversen C, Wendt D, Alickovic E, Whitmer WM, Lunner T. An exploratory Study of EEG Alpha Oscillation and Pupil Dilation in Hearing-Aid Users During Effortful listening to Continuous Speech. PLoS One. 2020 Jul 10;15(7):e0235782. doi: 10.1371/journal.pone.0235782.

Micula A, Rönnberg J, Fiedler L, Wendt D, Jørgensen MC, Larsen DK, Ng EHN. The Effects of Task Difficulty Predictability and Noise Reduction on Recall Performance and Pupil Dilation Responses. Ear Hear. 2021 Nov-Dec 01;42(6):1668-1679. doi: 10.1097/AUD.0000000000001053.

Fiedler L, Seifi Ala T, Graversen C, Alickovic E, Lunner T, Wendt D. Hearing Aid Noise Reduction Lowers the Sustained Listening Effort During Continuous Speech in Noise-A Combined Pupillometry and EEG Study. Ear Hear. 2021 Nov-Dec 01;42(6):1590-1601. doi: 10.1097/AUD.0000000000001050.

>> We have greatly revised the Introduction with your comments in mind and have added the suggested citations.

Methods:

The study group is too small. It should be minimum of 20 subjects in each subgroup (It would be the best to have 40-50 subjects). For some analyses only 15 subjects were taken. The authors should at least perform power analysis indicating what number of the subjects would be required for establishing significance.

>> We have added a power analysis and discussion in part 4.4. Limits to study.

Please describe briefly the whole procedure. Which tests where done first, which next? Were all tests made at the same session? How long took all the tests? Did the subjects got tired after all the testing? Was the problem of falling them asleep? How the authors tried to eliminate this factor?

>> We have added: “All tests were conducted in a single session which lasted an average of 65 minutes for the NH group and 100 minutes for the HA group, where tests were conducted with and without their HAs. The order of testing was EEG recording (15 minutes), followed by pupillometry (15 minutes), and then loudness rating (5 minutes). Between EEG recording and pupillometry, participants were given a 30-minute break. If a participant felt asleep during recording, the test was stopped and a break was given before continuing the session.”

In which program other analyses than FCPA were done? Also in R?

 >> FCPA was performed using R. We have added “Analyses were performed using IBM SPSS software.”

Results:

I suspect that for correlation analysis less than 20 subjects is way too small. I am afraid that some of the results might be incidental.

>> In the Limits to study section 4.4, we have added: “While this study demonstrated the impact of intensity on CAEPs and pupillometry, the number of participants was limited. Indeed, a power analysis with G*power software indicate that for power = 0.95, 43 participants would be needed for each group. Additional studies with a larger cohort should be conducted to confirm the present findings.”

All figures are very well prepared however the figure 6 has too small markers and captions.

>> Markers and captions have been enlarged for figure 6.

Discussion:

The first sentences of the discussion might be also the basis of providing better rationale in the Introduction.

“4.1. Effect of intensity level on loudness perception, CAEP and pupillometry” – there seem to be no need to mention Bonferroni correction in

>> We have removed the part of the sentence mentioning Bonferroni correction.

Conclusions:

Please consider rewriting or changing the title of this section to “Summary of the results” as there are no real conclusions here except that “Further research is needed to…” which is only an obvious general statement.

>> Changed as suggested.

Editing errors:

Too large font for section title “2.3.1. Categorical loudness scaling”.

>> The font was corrected.

Hard to distinguish subsection titles for “Cortical auditory evoked potentials (CAEPs)”: “Stimuli” and “Electroencephalogram (EEG) data recording”

>> We added spaces and italicize title.

Pages 6-7 – different font sizes throughout paragraphs.

>> Font sizes were corrected

“4.4. Limits to study” – change “unlear” to “unclear”

>> Corrected.

Reviewer 2 Report

This is a well written paper on the relationship between behavioral and objective measures of sound intensity in hearing aid users. It will potentially have a big impact on hearing aid fitting.

I have the following comments:

Page 5, Section 2.5 Analysis: why the mean pupil diameter was calculated within in a 5s time window after the stimulus onset? Is it more accurate using recordings of 2 sec after the onset of the stimulus as indicated in Fig. 4?

Page 10: HA group with CI on or off. Is this supposed to be with HA on or off? There are a couple more places using "CI" as well.

Author Response

This is a well written paper on the relationship between behavioral and objective measures of sound intensity in hearing aid users. It will potentially have a big impact on hearing aid fitting.

I have the following comments:

Page 5, Section 2.5 Analysis: why the mean pupil diameter was calculated within in a 5s time window after the stimulus onset? Is it more accurate using recordings of 2 sec after the onset of the stimulus as indicated in Fig. 4?

>> Sorry for the confusion. Fig. 4 shows that the tone burst was presented after the gray image was presented, after which there was a 20-30 s silent period. We have modified the relevant sentence: “The mean pupil response at each intensity presentation was estimate during a period of 5 s from baseline (1s before to 4s after stimulus onset).

Page 10: HA group with CI on or off. Is this supposed to be with HA on or off? There are a couple more places using "CI" as well.

>> Corrected throughout.

Reviewer 3 Report

The aim of the article to determine objective markers to describe the auditory comfort or discomfort and to compare behavioral and objective measures to vary the sound intensity is very interesting and innovative. CAEPs and pupillometry are well described such as the statistical investigation performed.
The tables help in this sense to make the data obtained more usable although they remain very complex. We suggest to implement the sample of patients and possibly evaluate the possibility of unifying the group by age since between HA and NH group there is a difference of about 12 years and could in our opinion affect the neural responses, according to aging of the brain structures.

Author Response

Comments and Suggestions for Authors

The aim of the article to determine objective markers to describe the auditory comfort or discomfort and to compare behavioral and objective measures to vary the sound intensity is very interesting and innovative. CAEPs and pupillometry are well described such as the statistical investigation performed.
The tables help in this sense to make the data obtained more usable although they remain very complex. We suggest to implement the sample of patients and possibly evaluate the possibility of unifying the group by age since between HA and NH group there is a difference of about 12 years and could in our opinion affect the neural responses, according to aging of the brain structures.

>> The main goal of the study was to analyze the impact of intensity on behavioral and objective measures determine possible markers of auditory comfort. This objective depended on analyzing of group data separately, Across-group comparisons were performed to complete analyses, but weren’t related to our main objective. Within the HA group, age would not be a factor for comparisons with the HA on or off.  When we compare NH with HA group, age could indeed influence responses. However, in many cases, parametric analyses could not be used because of violations to assumptions of normal data distribution and equal variance. Accordingly, we used non-parametric Kruskal Wallis ANOVAs on ranked data, where age could not be included as a covariable or factor.

We have added to section 4.3 of the Discussion: “Age might differentially affect different components of the pupil response reflecting para-sympathetic versus sympathetic effects [68]. In this study, the mean age of the NH group (64.0 ± 3.9 years) was 12 years younger than that of the HA group (76.0 ± 7.0 years). The older age of the HA group may partly explain the lack of association between loudness and pupil response, due to the reduced dynamic range of pupil dilation.” And later in section 4.4: “Age at testing was significantly different between both the NH and HA groups, and might have contributed to group differences by affecting neural responses. However, it is difficult to recruit adults up to 70 years old that still have normal hearing thresholds due to presbycusis.”

Round 2

Reviewer 1 Report

Thank you for addressing my comments.

Fig. 3 – please add x and y axis labels.

Author Response

Thank you for addressing my comments.

Fig. 3 – please add x and y axis labels.

>>  Thank you for your helpful comments. We add x and y axis labels on Figure 3.

Reviewer 3 Report

the corrections are right and I think that the manuscript is ready for the publication

Author Response

The corrections are right and I think that the manuscript is ready for the publication

>> Thank you for your helpful comments.
